# Development of an Instrument to Evaluate the Intake of Liquids, Food and Supplements in Endurance Competitions: Nutritional Intake Questionnaire for Endurance Competitions—NIQEC

**DOI:** 10.3390/nu15081969

**Published:** 2023-04-19

**Authors:** Rubén Jiménez-Alfageme, Mª Emilia Campodónico, Isabel Sospedra, Daniel Giménez-Monzo, Miguel García-Jaén, Rocío Juliá-Sanchís, Eva Ausó, José Miguel Martínez-Sanz

**Affiliations:** 1Faculty of Health Sciences, University of Alicante, 03690 Alicante, Spain; rja10@gcloud.ua.es (R.J.-A.); licmecampodonico@gmail.com (M.E.C.); 2Food and Nutrition Research Group (ALINUT), University of Alicante, 03690 Alicante, Spain; dgimenez@ua.es (D.G.-M.);; 3Physiotherapy Department, Faculty of Health Sciences, European University of Gasteiz—EUNEIZ, 01013 Vitoria-Gasteiz, Spain; 4Nursing Department, Faculty of Health Sciences, University of Alicante, 03690 Alicante, Spain; 5Department of Community Nursing, Preventive Medicine and Public Health and History of Science Health, University of Alicante, 03690 Alicante, Spain; 6Area of Physical Education and Sports, Faculty of Education, University of Alicante, 03690 San Vicente del Raspeig, Spain; m.garciajaen@ua.es; 7Department of Optics, Pharmacology and Anatomy, Faculty of Sciences, University of Alicante, 03690 Alicante, Spain

**Keywords:** questionnaire, endurance sports, nutritional intake, gastrointestinal complaints, development

## Abstract

Background: In the last few years endurance sports have experienced a great increase in the number of competitions and participants. Dietary-nutritional planning is key for performing well during such competitions. To date, there is no questionnaire expressly developed to be able to analyze the consumption of liquids, foods, and supplements, as well as gastrointestinal problems in these events. This study describes the development of the Nutritional Intake Questionnaire for Endurance Competitions (NIQEC). Methods: The study was composed in the following phases: (1) Bibliographic search for the most important nutrients, (2) focus groups (17 dietitian-nutritionists and 15 experienced athletes) and generation of items, (3) Delphi surveys, and (4) cognitive interviews. Results: After an initial shaping of the questionnaire with the items that emerged in the focus groups, their relevance was evaluated by means of the Delphi survey, which showed more than 80% approval for most items. Finally, the cognitive interviews indicated that the questionnaire was simple and complete for its purpose. The final NIQEC (*n* = 50 items) was divided in 5 sections: Demographic data; sports data; consumption of liquids, food and supplements before, during, and after the competition; gastrointestinal complaints, and dietary-nutritional planning for the competition. Conclusions: The NICEQ is a useful tool that allows collecting information from participants on sociodemographic factors and gastrointestinal complaints, and estimating the intake of liquid, food, and supplements, for endurance competitions.

## 1. Introduction

In recent years, endurance sports have grown in popularity, and an increasing number of athletes are competing at amateur, recreational, or elite levels, in competitions such as marathons or mountain races, or combined event competitions such as triathlons, duathlons, or aquathlons, where the physical effort can last from 2 h to 17 h, even exceeding 24 h [1].

Dietary-nutritional planning is important for having an adequate diet on the days before the competition, during and after the event, considering the characteristics of each discipline, event, and athlete, to enable optimal performance [2]. Regarding the main nutrients that should be ingested in this type of sporting event, various bibliographic reviews and meta-analyses [3,4,5] have found that a high consumption of carbohydrates (CHO) promotes better performance in this type of event [6]. The intake of fluids and sodium is also particularly important, due to their physiological importance and the prevention of hyponatremia [7]. To achieve these recommendations, endurance athletes, in addition to food and liquids, usually take sports supplements (SS), as previous studies have described [8,9,10]. In addition, gastrointestinal problems are a great challenge for both athletes and dietitian-nutritionists (D-N). However, the gastrointestinal tract is highly adaptable and specifically training it can improve nutrient delivery during exercise [11]. This training produces relief from the symptoms of discomfort such as abdominal distension, cramps, diarrhea or vomiting, which are a high cause of abandonment among endurance athletes [7,12].

For this reason, it is important to study in detail the types of diet consumed by athletes, as well as the incidence of gastrointestinal problems before, during, and after endurance competitions, in order to assess whether they comply with current scientific recommendations, and to establish future strategies, as previously done in other works [13,14]. However, as Masson and Lamarche [15] conclude in their work, a large part of endurance athletes do not know or reach the nutrient intake recommendations during this type of test, requiring more research on this aspect.

The studies found that collected intake data in endurance competitions are often based on the collection-by-interview method using trained D-Ns [16,17,18], as well as questionnaires prepared by specific research teams for certain competitions [13,14,19], which are sometimes based on modified 24 h intake reminders [20]. Other studies have developed questionnaires that measure the quality of the athlete’s diet [21] or their knowledge about nutrition [22], but to date, no questionnaires have been found in the literature that were developed and validated specifically to determine the dietary intake of liquids, food, and supplements associated with a specific competition, as well as the appearance of gastrointestinal symptoms. In the absence of a complete tool that integrates the eating behavior of athletes, with respect to liquids, foods, and supplements before, during, and after resistance sports competitions, as well as the incidence of gastrointestinal problems and their relationship with the consumption of a specific nutrient, food, or supplement, the proposed questionnaire will be a very useful tool and will facilitate research in this population in a precise, agile, and economical way.

Therefore, the aim of this study is to develop, through qualitative methodology that ensures content validity, a questionnaire to quantify the previously described intakes, as well as the occurrence of gastrointestinal discomfort in endurance athletes during competitions. Accordingly, it was initially hypothesized that it is possible to develop a questionnaire that takes into account different dimensions related to the athlete, dietary-nutritional preparation, as well as adverse outcomes and athletes’ fluid, food and supplement intake for endurance competitions.

## 2. Materials and Methods

### 2.1. Design

This is an instrumental study designed to develop a self-assessment tool [23] to assess fluid, food, and supplement intake, and the occurrence of gastrointestinal complaints an hour before, during, and an hour after endurance sports competitions.

The design and validation of the content of the questionnaire were developed in the following phases: (1) bibliographic review; (2) focus groups; (3) Delphi Survey; and (4) cognitive interviews. The items of the Delphi questionnaires were extracted from the bibliographic review focused on the purpose of the study, and further developed through focus group discussions. Subsequently, cognitive interviews were conducted to assess the comprehensibility and feasibility of the final questionnaire.

### 2.2. Procedures

#### 2.2.1. Phase 1: Bibliographic Search: Nutrients Selection

A review was carried out of the position and review documents of the leading scientific institutions in sports nutrition (American Academy of Nutrition and Dietetics, International Society for Sports Nutrition, the International Olympic Committee, and the Australian Institute of Sports) [24,25,26,27,28], to find the key nutrients in endurance competitions, their intake recommendations, and the main adverse results related to nutrition. Likewise, the theoretical sections and items that the questionnaire should include were identified. The results of this search constituted the theoretical basis for the preparation of a list of items grouped by topic.

#### 2.2.2. Phase 2: Focus Groups and Generation of Items

##### Participants

Four focus groups were held to deeply explore the consumption of liquids, food, and supplements an hour before, during, and an hour after the competition, as well as the incidence of gastrointestinal problems and their relationship with the consumption of a specific nutrient, food or supplement.

A minimum of 24 participants distributed by interest group was estimated to be appropriate. Two focus groups were composed of accredited dietitian-nutritionists with at least 2 years of work experience in the field of study, and another two groups were composed of federated athletes who engaged in endurance competitions and had at least 2 years of experience in the practice of and competition in these sports events.

The participants were selected through non-probabilistic intentional sampling, seeking representativeness by gender and group. The dietitian-nutritionists were identified through the Food and Nutrition Research Group (ALINUT) of the University of Alicante and the professional associations of dietitian-nutritionists in Spain. The athletes were identified through the dietitian-nutritionists. Two researchers (JMM-S and ISL) contacted the potential participants by email with a formal invitation to participate in this study, providing information on the objectives and characteristics of the research, as well as the methodology to be used and the type of assessment instrument for data collection. The final sample size was 32 participants: 17 dietitian-nutritionists and 15 experienced athletes, whose characteristics can be found in the Appendix A. With this, the theoretical saturation of the main categories and codes identified was reached.

##### Data Collection

A semi-structured script was used that addressed the key contents that the instrument should contain and the items it should include. Items related to the main topics extracted from the bibliographic search carried out in the previous phase included the sociodemographic and sporting data of the athlete, fluid intake, food and supplements an hour before, during and an hour after the competition, possible gastrointestinal problems, and planning for the competition at the dietary-nutritional level.

The sessions with the focus groups lasted an estimated 60 min, and were carried out online using the Microsoft Teams platform, where an environment of adequate intimacy was created. Focus groups were facilitated by two moderators (R.J.-A. or M.E.C.) and an assistant observer (M.G.-J.) with experience in qualitative research, who took notes. The focus groups were recorded in digital video format with prior consent from the participants, and later transcribed in text format, guaranteeing the anonymity of the participants.

##### Data Analysis

An inductive approach based on grounded theory was used [29]; this allows us to establish, through the organization of the data, the emerging theory on the subject under study. The analysis is based on procedures of description, conceptual ordering and theorizing. Coding was performed manually by three researchers with previous experience in qualitative analysis [30] (R.J.-A., M.E.C. and M.G.-J.). The resulting code map and its definitions, and a consistency chart between the research questions and the inferred codes, can be found in Appendix A.

The preliminary pool was created based on the qualitative data obtained from the bibliographic search and the focus groups, seeking exhaustiveness of the topics and maximum coverage of the content domains. The number of items was reduced by eliminating redundant, inconsistent, too specific, or generic items. The frequency of appearance (%FA) was calculated to assess the permanence or withdrawal of the items, with a total %FA < 50% leading to item withdrawal, while a total 50–80%FA was used for the theoretical discussion by the research team; ≥80%FA was a permanence criterion.

#### 2.2.3. Phase 3: Delphi Survey: Internal Validation

The Delphi technique in health sciences is commonly used to create measurement tools and identify indicators; it is a structured process of group communication in which experts evaluate, through an iterative process, complex issues in which knowledge is uncertain and incomplete [31].

For this study, a group of experts, the D-N and athletes (*n* = 32; Appendix A) who participated in the focus groups in Phase 1 were invited to review and evaluate (online) the structure of the questionnaire and the preliminary pool of items [32].

It is worth noting the importance of cognitive diversity in the composition of the panel of experts for the integrity and validity of the findings [31]. The participants were asked to indicate, for each item, whether it should be kept, modified, or excluded, by marking the degree of relevance using a Likert scale from 1 to 4 (1 being not very relevant and 4 being very relevant), together with an option to provide comments. Subsequently, the team of researchers, taking into account the quantitative scores and the qualitative contributions obtained in the first round of Delphi consultation, eliminated the items whose scores were equal to or less than 2 for at least 70% of the panelists, and whose median was less than 2.

Regarding the set of items that were initially selected, in a second round, the experts were asked about the adequacy of the items and whether they should be kept in the final version, marking yes or no. An adequate value was considered if the item obtained support greater than or equal to 80% [33].

#### 2.2.4. Phase 4: Cognitive Interviews

Finally, to assess the comprehensibility and feasibility of the final questionnaire, the items, the instructions, and the response format, cognitive interviews were conducted and recorded. The inclusion criteria for the participants in this validation process were to have spent at least 2 years practicing and competing in endurance competitions and having a native level of Spanish. All the participants voluntarily signed their informed consent. Thirteen cognitive interviews were carried out in person at the University of Alicante; 15% of participants were women, the participants’ mean age was 24.1 ± 3.98 years, and all were Spanish.

The interviews were conducted by researchers trained in the process of creating and adapting instruments (R.J.-S., J.M.M.-S. and R.J.-A.). First, each part of the preliminary version of the instrument in Spanish (instructions, response options, and items) was read aloud. The participants were asked to indicate: (a) general understanding of each part; (b) difficulty understanding any term; and (c) the degree of difficulty of each section by choosing two response options: 1—very clear, simple, and easy to understand; and 2—very complex and difficult to understand. In the case of answering the second option, the participant was asked to propose an alternative expression that would make it easier for him or her to understand. Finally, they were asked to make a general assessment of the questionnaire.

Based on the suggested changes, the researchers modified the questionnaire, resulting in the final version. A descriptive analysis of all items was performed using the mean ± standard deviation.

### 2.3. Ethical Considerations

The study complied with the Declaration of Helsinki for research in humans and was approved by the Ethics Committee of the University of Alicante (UA File 2021-02-01).

## 3. Results

### 3.1. Phase 1: Bibliographic Search

After reviewing the previously described literature, carbohydrates, sugars, proteins, and sodium were determined as key nutrients for this type of competition, all of them considered grade A evidence [24,25,26,27,28]. According to the evidence, the amount of kcal consumed, fats, water, and other supplements such as caffeine, were elements of interest in endurance sports competitions [26,34].

### 3.2. Phase 2: Focus Groups and Item Generation

After the qualitative analysis of the data extracted from the meetings with the 4 focus groups, the topics that composed the different parts of the questionnaire were constituted, as well as the coding of the narrative units expressed by the participants in the focus groups. The results derived from the coding generated the categories of meaning from which the necessary items that must be included in the questionnaire were identified, as well as their corresponding frequencies of appearance, expressed both in absolute values and as a percentage of the total number of participants in each of the focus groups.

The results from the focus groups are presented in Table 1, Table 2, Table 3, Table 4 and Table 5. These show the main topics that make up the different parts in which the questionnaire is to be structured, the codes and items identified as necessary by the participants during the meetings, and the frequency of appearance (FA) that allows assessing the importance of each item within the questionnaire. The main topics proposed included the athlete’s demographic data; sports data; the consumption of food, liquids and supplements before, during and after the competition; the possible gastrointestinal problems derived from the said consumption; the dietary-nutritional planning of the athlete who will compete. In the item generation tables (Table 1, Table 2 and Table 3), the topics and categories that emerged during the meetings were coded.

Regarding the results from this metacoding, it can be observed that most of the codes, both from the point of view of the D-N and the athletes, obtained a similar percentage of frequency of appearance (%FA), greater than 60%. However, for A1.2 and A1.3, the codes related to age (47.05%) and date of birth (52.94%), respectively, the %FA was lower than the rest, due to the overlap between these codes, as both referred to knowing the age of the athlete. It may be useful to ask both the age and the date of birth. According to the opinion of the D-Ns: “To make it easier, when asking for the date of birth, ask it directly” (Participant D-N, number 3); “To avoid variability in the response, “Age (years)” could be indicated and the response will be continuous quantitative” (Participant D-N, number 6); and “I would put the date of birth to have a more exact decimal age” (Participant D-N, number 15). The principal investigators decided to include both age and date of birth in order to more accurately perform the nutritional calculation and description of the sample, as the date of birth allows obtaining the exact age in decimal format.

In relation to the second topic of the athlete’s sports data, we find the metacodes B1 and B2 (questions referring to sports data from the point of view of the D-N and the athletes), with some variation in %FA (Table 2).

Regarding the results that emerged from the competition level metacodes (B1.1, B2.1), both groups suggested modifications to the question with comments such as: “I would add the modalities duathlon, marathon, etc. People who have been practicing endurance sports for many years can specialize in one for a few years, and in others in other years” (Athlete participant, number 12); “No, on many occasions several disciplines can be combined” (Participant D-N, number 8); “I would ask what sport you do; many mountain bikers train mostly with road cycling, even though they compete in mountain biking, so the question can be a bit biased in order to personalize later.” (Participant D-N, number 13); “The options could be added: “Asphalt races (MM, marathon)” and Others (Mountain biking, aquathlon, adventure, etc.)” (Participant D-N, number 6); “It would be convenient to specify more, for example, the type of triathlon (sprint, super sprint…), half marathon etc.” (Athlete participant, number 14). In the rest of the cases, the %FA was greater than 50%.

Table 3 presents the results for the topic “Consumption of liquids, food and supplements an hour before, during, and in the hour after the competition”. The C1 metacode related to the consumption of food, liquids and supplements 1 h before the competition from the point of view of the D-N, and metacode C4 covers the same topic from the point of view of the athletes; metacodes C2 and C5 related to the consumption of liquid food and supplements during the competition from the point of view of D-N and athletes, respectively. Finally, metacodes C3 and C6 related to the point of view of D-N and athletes on the consumption of liquid food and supplements in the hour after the competition (Table 3a,b).

The results for this topic showed that the %FA were greater than 50% in all cases.

For issues related to “during” the competition, data on subcodes were obtained within the Solid Foods code C2.2: C2.2.1 fruits, C2.2.2 nuts, C2.2.3 homemade energy bars, C2.2.4 commercial energy bars, C2.2.5 others, with total %FA for all subcodes of 88.23% in the case of D-N experts, and 80% from the point of view of athletes (C5.1.1, C5.1.2, C5.1.3, C5.1.4, C5.1.5). For liquids, data were obtained for the following subcodes: C2.7.1 water, C2.7.2 soft drinks, C2.7.3 coffee/Infusions, C2.7.4 broths/soups, C2.7.5 replacement drinks (isotonic or sports drinks), with a total %FA of 91.76% from the point of view of the D-N, and a total of 86.67% from the point of view of the athletes (C5.7.1, C5.7.2, C5.7.3, C5.7.4, C5.7.5).

The fourth topic related to gastrointestinal problems, with the metacodes D1 and D2 covering the point of view of D-N and athletes, respectively (Table 4).

It was observed that all the codes obtained a %FA greater than or equal to 80%, indicating the importance of these items within the questionnaire, given the aim of assessing the relationship between endurance competition and the gastrointestinal problems that may be associated with it. Comments included “Correct (ask if they have performed the training of the digestive system) (Athlete participant, number 10)”; “Correct. In the case of a triathlon, I would also ask the segment in which the athlete has suffered, since it is usually much more common in the race due to the impact itself, which increases the risk (Participant D-N, number 11)”; and “Correct, I really like that the answer includes the symptoms and that I can choose, since on many occasions they don’t even know how to identify them (Participant D-N, number 8)”.

Finally, Table 5 shows the metacodes that emerged for the topic of nutritional dietary planning for competitions from the point of view of the D-N experts and athletes (E1, E2).

The results for this last topic showed us that in all the codes, the %FA was greater than 50% both from the point of view of the D-N experts and the athletes. Hence, this is an important topic to include in the questionnaire in order to obtain information about the nutritional dietary planning of the athlete for his or her competition.

### 3.3. Phase 3: Delphi Survey: Internal Validation

The D-Ns and athletes (*n* = 32) who participated in the focus groups in Phase 1 were invited to participate in the Delphi survey, with all of them responding positively to the said invitation. The results in relation to each topic were perceived as relevant by answering Y (Yes, definitively include this question) and as not relevant by answering with N (Do not include in the final questionnaire or reformulate the questions explaining the reason) (Table 6). Most of the items included within each topic were valued as relevant with percentages higher than 80%. In the case of the questions considered not relevant, they represented a total of less than 20% in each topic.

### 3.4. Phase 4: Cognitive Interviews

In general, the interviewees indicated that the scale was “simple” and “complete” for the evaluation of the consumption of liquids, food and supplements in the hour before, during and in the hour after the competition, as well as the incidence of gastrointestinal problems. All the participants adequately understood the meaning of the instructions, response options, and items. Furthermore, they agreed with the content and did not suggest the addition of any new item. However, five of the sixteen items were modified because they were difficult to understand. This last phase made it possible to define and internally validate the defined questionnaire (instrument) to achieve its objective (Appendix A). The instrument was named the Nutritional Intake Questionnaire for Endurance Competitions (NIQEC) (see Appendix A).

## 4. Discussion

The results of this study through the use of qualitative methodology, support the content and show the content validity of the NICEQ and its perceived relevance, as well as its usefulness for athletes. The final NICEQ includes 50 items divided into demographic data, sports data, food, fluid, and supplement intake before, during and after the sporting event, as well as gastrointestinal problems and competition planning at the dietary-nutritional level. The NICEQ concept was valued positively by both athletes and D-N experts.

The methodology followed in the elaboration of this questionnaire was similar to the work carried out by Capling et al. [21]. However, their work assessed the general quality of the diet of athletes, unlike the NIQEC which assesses the diet of athletes during endurance competitions. In this sense, and as mentioned in the study by Villanova et al. [35], the questionnaire must unify methodological criteria, and if the desired aim is to assess the consumption of sports supplements in the athlete population who compete, it should include: sociodemographic factors, sports practice (sports modality or discipline practiced by the subjects to whom the questionnaire is addressed), and reasons/motives for use and consumption, aspects that were taken into consideration in the development of this questionnaire.

Part of the questionnaire was designed to collect individual information about the consumption of food and supplements before, during and after the endurance competitions, with these data later utilized for performing a nutritional analysis to estimate the amount of water, sodium, carbohydrates, and protein consumed, which allows a comparison with current recommendations for improving sports performance. In this sense, nutritional recommendations have been established of an average hourly intake during competition of 400–800 mL of liquid, 300–600 mg of sodium, and 30–120 g of CHO [24,25,26,36,37]. This quantity depends on the type of sport and competition, individual characteristics of athletes, orography, and competition profile, as well as other factors such as environmental conditions, e.g., humidity and temperature [26,38]. In addition to this, the post-exercise nutritional recovery is contemplated through the intake of carbohydrates and proteins (0.8 g CHO/kg body weight plus 0.2–0.4 g protein/kg body weight) [24,25,26,39].

Regarding the evaluation of food and supplement consumption with respect to competitions, according to the existing bibliography, we can find various methods to evaluate dietary-nutritional intake [40], with the choice of a particular method dependent on the objectives and available resources, but they are mostly based on food frequency questionnaires, 24-h reminders or dietary records [40]. These direct dietary assessment methods aim to obtain the habitual and/or current intake of individuals, and some of their limitations are the need for trained interviewers to collect the information, the complexity of completing the data collection, and the need for the methods to be validated in the population where they will be used [41]. Moreover, available studies do not provide specific information on pre/per/post competition intake. Some studies have analyzed the dietary-nutritional intake in endurance athletes during competitions through interviews with trained D-Ns [16,17,18], while other studies have used competition-specific questionnaires developed by the research teams themselves [13], or are based on data obtained from modified 24-h recalls [20].

All of this together highlights the existing problems when collecting quantitative information on the intake of food and supplements in the athlete population, as well as the need to validate dietary evaluation methods that can help with the quantification of portions, the reduction in the burden of data collection, and the problems with the lack of recording of food consumption in the athlete population [21].

Other research studies, whose aim was to assess gastrointestinal problems associated with exercise, have used different types of questionnaires [13,42]. These allowed obtaining information based on a rating scale of the symptoms associated with the event, as in the NICEQ. Since the questionnaire provides specific information on the intake of water, food, and supplements at different points of the competitions, it could help to investigate the possible relationship between their consumption and gastrointestinal discomfort, with this being one of the main problems reported by athletes in endurance sports competitions [7,43].

There is no standardized methodology in the questionnaires in the scientific literature, and there is also a lack of homogeneity in the type and number of variables used to estimate the use and consumption of supplements during sporting events [35]. It is also important to note that, as indicated by the panel of experts consulted, none of the questionnaires used in previous studies made it possible to cover all the important topics [44] in the same questionnaire, in regard to questions that referred to the consumption of food, liquids, and supplements, as well as knowing the associated gastrointestinal problems, previous planning, and sports data associated with the event.

Therefore, in order to achieve a questionnaire that is as accurate as possible in achieving the objective proposed in this research, it is useful to follow a qualitative methodology for generating items, through focus groups of experts, and their subsequent evaluation using the Delphi methodology. Likewise, during the emergency health situation due to COVID-19 [45], it is interesting to utilize an online format resource, as it does not require direct contact with the athlete. It can be used to obtain the data quickly, easily, and safely, which also allows reaching a higher number of athletes and eliminates the need for trained interviewers to collect information. Due to these factors, the NICEQ is a very useful tool in the field of nutrition research in endurance sports.

Finally, it is interesting to note that some athletes must travel to the places where the resistance competitions will take place. In the case of traveling by plane, they may be affected by jet lag can occasionally have serious consequences for the athlete’s mental and physical health and performance [46,47]. This jet lag could produce acute dehydration which may affect the general health or performance of elite athletes [46] and which may also be produced by a bad heat acclimation [48]. These types of problem could be reflected in the results of the NICEQ.


**Study limitations**


Regarding the limitations of the study, it is known that the good use of questionnaires strongly depends on the memory of the athletes, which may constitute a challenge, especially with regard to providing correct estimates of fluid intake during prolonged races, and detailed instructions prior to the event will be necessary with strategies in place so that the athlete is able to remember the race intake. However, in general, competitive athletes tend to be very committed to every aspect of sports performance, which entails a better memory of their food intake around the competition period. Another limitation is that the present work carried out an internal validation through the panel of experts using a qualitative methodology. Despite the creation of this questionnaire in Spanish, future research will require different cross-cultural adjustments for work with Spanish speakers in other countries and contexts. Another limitation is that intraclass reliability (ICC) data have not been indicated to assess the consistency or reproducibility of measures made by different observers or by the same observer at other times. However, the study of the tool’s psychometric properties, validity and reliability will be presented in the next research phase through a pilot study in endurance competitions (exploratory factor analysis and confirmatory factor analysis). This new phase will make it possible to confirm the construct validity of the questionnaire.

## 5. Conclusions and Practical Application

The NICEQ is the first endurance-specific questionnaire to be developed. It is a useful tool that allows collecting information from participants on socio-demographic factors, gastrointestinal complaints, and sports practices, as well as estimating the intake of food and supplements before, during and after competitions in endurance sports.

In addition, the NICEQ will allow estimating, describing and comparing in different competitions: (1) the type of liquid, food or supplement consumed; (2) the nutritional intake of kcal, macronutrients (CHO, lipids and proteins), sodium and caffeine; (3) incidence and causes of gastrointestinal complaints and their relationship with food intake; (4) compliance with the dietary-nutritional recommendations for the hour before, during and the hour after the competition.

## Figures and Tables

**Table 1 nutrients-15-01969-t001:** Results of the focus groups for the content of part I of the questionnaire, related to the demographic data of the athlete.

Proposed Topics	Identified Topic	FA Code (FA %)
A1. Demographic data from the point of view of expert dietitian-nutritionists	A1.1 Sex	14 (82.4%)
	A1.2 Birthdate	8 (47.1%)
	A1.3 Age	9 (52.9%)
	A1.4 Autonomous community of residence	16 (94.1%)
	A1.5 Height	14 (82.4%)
	A1.6 Current weight	14 (82.4%)
A2. Demographic data from the point of view of athletes	A2.1 Sex	14 (93.3%)
	A2.2 Birthdate	1 (6.7%)
	A2.3 Age	14 (93.3%)
	A2.4 Autonomous community of residence	14 (93.3%)
	A2.5 Height	15 (100%)
	A2.6 Current weight	15 (100%)

**Table 2 nutrients-15-01969-t002:** Results of the focus groups for the content of part II of the questionnaire, related to sports data.

Proposed Topics	Identified Topic	FA Code (FA %)
B1. Sports data from the point of view of expert dietitian-nutritionists	B1.1 Main competitive sport	6 (35.29%)
	B1.2 Number of training sessions per week	12 (70.6%)
	B1.3 Performing 2 or more training sessions per day	10 (58.8%)
	B 1.4 Hours of training per week	14 (82.4%)
	B 1.5 Years you have been doing the sport	13 (76.5%)
	B1.6 Federated	17 (100%)
	B1.7 Competitive level	7 (41.2%)
	B1.8 Race number	9 (52.9%)
	B1.9 Distance of the event performed	10 (58.8%)
	B1.10 Event completion time	16 (94.1%)
	B1.11 In case you have not finished the event	15 (88.2)
B2. Sports data from the point of view of athletes	B2.1 Main competitive sport	7 (46.7%)
	B2.2 Number of training sessions per week	12 (80%
	B2.3 Performing 2 or more training sessions per day	13 (86.7%)
	B 2.4 Hours of training per week	13 (86.7%)
	B 2.5 Years you have been doing the sport	14 (93.3%)
	B2.6 Federated	15 (100%)
	B2.7 Competitive level	10 (66.7%)
	B2.8 Race number	8 (53.5%)
	B2.9 Distance of the event performed	11 (73.3%)
	B2.10 Event completion time	11 (73.3%)
	B2.11 In case you have not finished the event	13 (86.7%)

**Table 3 nutrients-15-01969-t003:** a. Results of the focus groups for the content of part III of the questionnaire, referring to the consumption of liquids, food and supplements one hour before, during and the hour after the competition from the point of view of expert dietitian-nutritionists. b. Results of the focus groups for the content of part III of the questionnaire, referring to the consumption of liquids, food and supplements one hour before, during and the hour after the competition from the point of view of the athletes.

Proposed Topics	Identified Topic	FA Code (FA %)
(a)—Point of view of expert dietitian-nutritionists		
C1. Consumption of food, liquids and supplements 1 h before the competition from the point of view of expert dietitian-nutritionists	C1.1 Solid food	11 (64.7%)
	C1.2 Liquids	15 (88.23%)
	C1.3 Supplements	15 (88.23%)
C2. Consumption of food, liquids and supplements during the competition from the point of view of expert dietitian-nutritionists	C2.1 Consumption of solid/semi-solid food during the competition	13 (76.5%)
	C2.2 Solid food	15 (88.23%)
	C2.3 Consumption of gels during the competition	17 (100%)
	C2.4 Quantity of gels consumed during the competition	17 (100%)
	C2.5 Type of gel	17 (100%)
	C2.6 Drinks consumption during the competition	17 (100%)
	C2.7 Drinks	16 (94.1%)
	C2.8 Consumption of some type of supplement not included in the previous questions during the test? (salt pills, caffeine pills, etc…).	17 (100%)
	C2.9 Types of supplement	17 (100%)
	C2.10 Details of types of supplement	15 (88.23%)
C3. Consumption of food, liquids and supplements 1 h after the competition from the point of view of expert dietitian-nutritionists	C3.1 Solid food	17 (100%)
	C3.2 Liquids	15 (88.23%)
	C3.3 Supplements	15 (88.23%)
(b)—Point of view of the athletes		
C4. Consumption of food, liquids and supplements 1 h before the competition from the point of view of the athletes	C4.1 Solid food	13 (86.7%)
	C4.2 Liquids	13 (86.7%)
	C4.3 Supplements	13 (86.7%)
C5. Consumption of food, liquids and supplements during the competition from the point of view of the athletes	C5.1 Consumption of solid/semi-solid food during the competition	13 (86.7%)
	C5.2 Solid food	11 (73.3%)
	C5.3 Consumption of gels during the competition	14 (93.3%)
	C5.4 Quantity of gels consumed during the competition	13 (86.7%)
	C5.5 Type of gel	12 (80%)
	C5.6 Drinks consumption during the competition	13 (86.7%)
	C5.7 Drinks	13 (86.7%)
	C5.8 Consumption of some type of supplement not included in the previous questions during the test? (salt pills, caffeine pills, etc…).	13 (86.7%)
	C5.9 Types of supplement	13 (86.7%)
	C5.10 Details of types of supplement	14 (93.3%)
C6. Consumption of food, liquids and supplements 1 h after the competition from the point of view of the athletes	C6.1 Solid food	14 (93.3%)
	C6.2 Liquids	13 (86.7%)
	C6.3 Supplements	13 (86.7%)

**Table 4 nutrients-15-01969-t004:** Results of the focus groups for the content of part IV of the questionnaire, related to gastrointestinal problems.

Proposed Topics	Identified Topic	FA Code (FA %)
D1. Gastrointestinal complaints from the point of view of expert dietitian-nutritionists	D1.1 Diagnosis of any food allergy or intolerance	15 (88.2%)
	D1.2 Action to take in the affirmative case of allergy or food intolerance	16 (94.1%)
	D1.3 Gastrointestinal discomfort or complaint during the competition	16 (94.1%)
	D1.4 Type of discomfort with its degree of intensity	14 (82.4%)
	D1.5 Gastrointestinal discomfort/problem during the competition is related to the consumption of any food, liquid or supplement	16 (94.1%)
	1.6 In case of discomfort, the type of food that causes it	14 (82.4%)
	D1.7 Reason or reasons why it could have happened	14 (82.4%)
D2. Gastrointestinal complaints from the point of view of athletes	D2.1 Diagnosis of any food allergy or intolerance	13 (86.7%)
	D2.2 Action to take in the affirmative case of allergy or food intolerance	12 (80%)
	D2.3 Gastrointestinal discomfort or complaint during the competition	13 (86.7%)
	D2.4 Type of discomfort with its degree of intensity	12 (80%)
	D2.5 Gastrointestinal discomfort/problem during the competition is related to the consumption of any food, liquid or supplement	14 (93.3%)
	D2.6 In case of discomfort, the type of food that causes it	13 (86.7%)
	D2.7 Reason or reasons why it could have happened	12 (80%)

**Table 5 nutrients-15-01969-t005:** Results of the focus groups for the content of part V of the questionnaire, related to the competition dietary-nutritional planning.

Proposed Topics	Identified Topic	FA Code (FA %)
E1. Planning of the dietary-nutritional actions from the point of view of expert dietitian-nutritionists	E1.1 Special preparation (training sessions for the gastrointestinal tract, specific diet days before, etc…) for this competition	14 (82.4%)
	E.1.2 Previous planning of the dietary-nutritional strategy for the competition	17 (100%)
	E1.3 Nutritional advice received from a professional before the competition	10 (58.8%)
E2. Planning of the dietary-nutritional actions from the point of view of athletes	E2.1 Special preparation (training sessions for the gastrointestinal tract, specific diet days before, etc…) for this competition	14 (93.3%)
	E.2.2 Previous planning of the dietary-nutritional strategy for the competition	15 (100%)
	E2.3 Nutritional advice received from a professional before the competition	13 (86.7%)

**Table 6 nutrients-15-01969-t006:** D-N and athletes’ responses to the Delphi Survey.

Topics	Very Relevant	Not Very Relevant
Demographic data	95 (88%)	13 (12%)
Sport data	178 (89.9%)	20 (10.1%)
Intake of food, liquids and supplements before, during, and after the event	402 (97.1%)	12 (2.9%)
Gastrointestinal complaints	106 (98.1%)	2 (1.9%)
Planning for the competition at the dietary-nutritional level	57 (100%)	0

## Data Availability

The data from this study are available on request from the corresponding author.

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
