# Peer review of "Development of an Instrument to Evaluate the Intake of Liquids, Food and Supplements in Endurance Competitions: Nutritional Intake Questionnaire for Endurance Competitions—NIQEC"

_nutrients, 2023, doi:10.3390/nu15081969_

Round 1

Reviewer 1 Report

Thank you for the opportunity to revise the current MS. Please find my comments and suggestions below, line by line:

Abstract:

Line 23: Typo, please amend.

Introduction:

What if these endurance competition require long distance travel and are influenced by jet lag for example ? Please see the following

Up in the Air: Evidence of Dehydration Risk and Long-Haul Flight on Athletic Performance - PubMed (nih.gov)

The elite athlete as a special risk traveler and the jet lag's effect: lessons learned from the past and how to be prepared for the next Olympic Games 2020 Tokyo - PubMed (nih.gov)

Impact of Long-Haul Travel to International Competition on Sleep and Recovery in Elite Male and Female Soccer Athletes - PubMed (nih.gov)

Is there any study hypothesis you might outline here ?

Methods:

Line 89: What does this mean? Instrumental study design is a bit unclear

Lines 114 – 118: Why is 2 years of experience established as minimal required threshold

Line 144: This allows…..

Line 288: Typo, please amend.

Results:

Are there any data on the intra assay reliability of the items you collected here ? This is quite important to show consistency of data collected. For me this a vital part of both Methods and Results.

Discussion

Line 374: Typo, please amend.

Please parts of the MS (lines 390-391), they seem a bit disconnected.

Conclusions

Couple sentences on practical insight from you work would be apricated here.

Author Response

Abstract:

Line 23: Typo, please amend.

Response of the authors: We appreciate the reviewer's comment, the typo has been fixed (comma deleted).

Introduction:

What if these endurance competition require long distance travel and are influenced by jet lag for example ? Please see the following

Up in the Air: Evidence of Dehydration Risk and Long-Haul Flight on Athletic Performance - PubMed (nih.gov)

The elite athlete as a special risk traveler and the jet lag's effect: lessons learned from the past and how to be prepared for the next Olympic Games 2020 Tokyo - PubMed (nih.gov)

Impact of Long-Haul Travel to International Competition on Sleep and Recovery in Elite Male and Female Soccer Athletes - PubMed (nih.gov)

Response of the authors: We appreciate the reviewer's comment and it is important to take into account the influence of jet lag. However, the objective of this questionnaire does not contemplate this dimension as we focus on the intake of the hour before, during and the hour after the competition.

It is true that jet lag and acclimatization can influence dietary-nutritional intake and the development of an Endurance competition, but this aspect could be reported in the questions “4. If you have not completed the competition, indicate the reasons” and “47. Explain the reasons why you think it could have happened.

In addition, and given the importance of this aspect, it has been included in the discussion section.

Is there any study hypothesis you might outline here ?

Response of the authors: We appreciate the reviewer's comment, the authors have added a hypothesis.

Methods:

Line 89: What does this mean? Instrumental study design is a bit unclear

Response of the authors: We appreciate the reviewer's comment, a research instrument study design refers to the "plan or strategy for selecting and using a particular research instrument or set of instruments to collect data" (Creswell, 2014). This type of design involves carefully selecting and testing various research instruments, such as surveys, interviews, or observation protocols, to ensure that they are valid and reliable measures of the variables being studied. Ultimately, the goal of a research instrument study design is to choose the most appropriate tool or combination of tools to collect data that will help answer the research questions at hand.

Creswell, J. W. (2014). Research Design: Qualitative, Quantitative and Mixed Methods Approaches (4th ed.). Thousand Oaks, CA: Sage.

Lines 114 – 118: Why is 2 years of experience established as minimal required threshold

 Response of the authors: We appreciate the reviewer's comment.

The research team agreed to establish this criterion to guarantee, on the one hand, that dietitians-nutritionists have experience in advising dietetic-nutritional athletes of resistance and know the problems or inconveniences at a nutritional level in resistance competitions.

On the other hand, it is very important that athletes have competed in different resistance competitions and have experienced the intake of liquids, food, and supplements during them, as well as the appearance of problems or inconveniences at the nutritional level.

This aspect can guarantee the suitability and certainty of the questions posed so that the questionnaire can adequately collect the information for which it has been developed.

Line 144: This allows…..

Response of the authors: We appreciate the reviewer's comment, the phrase has been rewritten for better understanding. 

Line 288: Typo, please amend.

Response of the authors: The typographical error has been corrected, removing the space

Results:

Are there any data on the intra assay reliability of the items you collected here ? This is quite important to show consistency of data collected. For me this a vital part of both Methods and Results.

Response of the authors: The present study aimed to develop, using qualitative methodology, a questionnaire to quantify the previously described intakes and the occurrence of gastrointestinal discomfort in endurance athletes during competitions. To this end, the design and validation of the content of the questionnaire were developed in the following phases: (1) literature review; (2) focus groups; (3) Delphi survey; and (4) cognitive interviews.

Like you, we will also find it relevant to note intraclass reliability (ICC) data. This statistical measure assesses the consistency or reproducibility of measures made by different or the same observer at other times. However, the study of the tool's psychometric properties, validity and reliability will be presented in future research. Given its importance, it is essential to note this as a limitation of our study.

Discussion:

Line 374: Typo, please amend.

Response of the authors: We appreciate the reviewer's comment, the typo has been fixed.

Please parts of the MS (lines 390-391), they seem a bit disconnected.

Response of the authors: We appreciate the reviewer's comment, the sentence has been modified.

Conclusions:

Couple sentences on practical insight from you work would be apricated here.

Response of the authors: We appreciate the reviewer's comment, this section has been modified in “Conclusions and practical application”.

Reviewer 2 Report

The current manuscript aims to develop an instrument by which nutritional intake (macronutrients, kcals, electrolytes, fluids, and supplements) can be assessed in a select population - endurance athletes. Moreover, this study seeks to examine nutritional parameters in the restricted time period closest to endurance competitions.

The questionnaire instrument developed here could provide useful insight into the nutritional intake of endurance athletes and track nutritional status in relation to performance, adverse events such as gastrointestinal distress, and perhaps recovery post-competition.

The manuscript is generally well-written, with some notes on improvements suggested below in this review. The text could be made more concise without loss of clarity by a thorough proofreading. The methods describe the approach to design, data collection, and validation with adequate detail. The use of qualitative research methods in the form of focus groups, interviews, and the Delphi method is warranted in a study of this type, which aims to develop a valid instrument to assess nutritional status in endurance athletes. Given the critical importance of proper peri-competition nutrition in endurance athletic events, an accurate nutrition assessment questionnaire could be expected to add valuable insight into the relationship between nutrition and performance. This could enable us to increase our understanding of best practices and refine nutritional recommendations for endurance athletes. The limitations of the study are well described by the authors.

A few specific suggestions:

- line 347: please rephrase. Use and usefulness are repetitive and awkward.

- 356: missing word after desired. effect? aim?

Author Response

The current manuscript aims to develop an instrument by which nutritional intake (macronutrients, kcals, electrolytes, fluids, and supplements) can be assessed in a select population - endurance athletes. Moreover, this study seeks to examine nutritional parameters in the restricted time period closest to endurance competitions.

The questionnaire instrument developed here could provide useful insight into the nutritional intake of endurance athletes and track nutritional status in relation to performance, adverse events such as gastrointestinal distress, and perhaps recovery post-competition.

The manuscript is generally well-written, with some notes on improvements suggested below in this review. The text could be made more concise without loss of clarity by a thorough proofreading. The methods describe the approach to design, data collection, and validation with adequate detail. The use of qualitative research methods in the form of focus groups, interviews, and the Delphi method is warranted in a study of this type, which aims to develop a valid instrument to assess nutritional status in endurance athletes. Given the critical importance of proper peri-competition nutrition in endurance athletic events, an accurate nutrition assessment questionnaire could be expected to add valuable insight into the relationship between nutrition and performance. This could enable us to increase our understanding of best practices and refine nutritional recommendations for endurance athletes. The limitations of the study are well described by the authors.

Response of the authors: We appreciate the reviewer's comments and thank the interest in the work done.

A few specific suggestions:

- line 347: please rephrase. Use and usefulness are repetitive and awkward.

Response of the authors: We appreciate the reviewer's comment, and the phrase has been rephrased to avoid that repetition and missense.

- 356: missing word after desired. effect? aim?

Response of the authors: We appreciate the reviewer's comment, the word “aim” has been included.

Reviewer 3 Report

I congratulate the authors! This study creates a fantastic, useful and relevant tool. This tool will have enormous applicability and will fill a current need. 

Author Response

We appreciate the reviewer's comments and thank the interest in the work done. 

We hope that soon the manuscript will be accepted and published to start using it in different endurance competitions.

We believe that it can provide valuable information on compliance or adequacy of dietary-nutritional recommendations for endurance events.

Round 2

Reviewer 1 Report

Thank you for following my comments and suggestions. For the final version of the MS please give this text to a native Eng. speaker. 

Author Response

Dear reviewer, thank you for your contributions and suggestions. We inform you that the manuscript has been reviewed by a native English speaker.